# Long-lived efficient delayed fluorescence organic light-emitting diodes using n-type hosts

Lin-Song Cui[1], Shi-Bin Ruan[1], Fatima Bencheikh[1,2], Ryo Nagata[1], Lei Zhang[3], Ko Inada[1], Hajime Nakanotani[1,2,4], Liang-Sheng Liao[3] & Chihaya Adachi[1,2,4]

Organic light-emitting diodes have become a mainstream display technology because of their desirable features. Third-generation electroluminescent devices that emit light through a mechanism called thermally activated delayed fluorescence are currently garnering much attention. However, unsatisfactory device stability is still an unresolved issue in this field. Here we demonstrate that electron-transporting n-type hosts, which typically include an acceptor moiety in their chemical structure, have the intrinsic ability to balance the charge fluxes and broaden the recombination zone in delayed fluorescence organic electroluminescent devices, while at the same time preventing the formation of high-energy excitons. The n-type hosts lengthen the lifetimes of green and blue delayed fluorescence devices by > 30 and 1000 times, respectively. Our results indicate that n-type hosts are suitable to realize stable delayed fluorescence organic electroluminescent devices.

[1] Center for Organic Photonics and Electronics Research (OPERA), Kyushu University, 744 Motooka, Nishi, Fukuoka 819-0395, Japan. [2] JST, ERATO, Adachi Molecular Exciton Engineering Project, 744 Motooka, Nishi, Fukuoka 819-0395, Japan. [3] Jiangsu Key Laboratory for Carbon-Based Functional Materials & Devices, Institute of Functional Nano & Soft Materials (FUNSOM) & Collaborative Innovation Center of Suzhou Nano Science and Technology, Soochow University, Suzhou, 215123, China. [4] International Institute for Carbon Neutral Energy Research (WPI-I2CNER), Kyushu University, 744 MotookaNishiFukuoka, 819-0395, Japan. Correspondence and requests for materials should be addressed to L.-S.L. (email: lsliao@suda.edu.cn) or to C.A. (email: adachi@cstf.kyushu-u.ac.jp)

Emitters exhibiting thermally activated delayed fluorescence (TADF) are promising as the third generation of luminescent materials for use in organic light-emitting diodes (OLEDs)[1,2]. In TADF molecules, triplet excitons ($T_1$), which generally non-radiatively decay before emitting light in traditional fluorescent materials, are readily up-converted into singlet excitons ($S_1$), which can easily emit light by fluorescence, because the $S_1$ and $T_1$ states are nearly degenerate, leading to 100% internal quantum efficiency[3]. As a result, TADF emitters have received considerable attention, and many efficient TADF molecules have been developed[4–10]. However, TADF technology still has some outstanding issues, such as the unsatisfactory stability of devices containing TADF emitters.

In a typical TADF OLED, holes and electrons injected from opposing electrodes travel through transport and blocking layers before meeting in the emission layer (EML), where they recombine to form excitons[11]. The EML usually consists of a wide-energy-gap host material doped with a TADF guest. This combination allows for efficient Förster and Dexter energy transfer from the host to the guest and confinement of both singlet and triplet excitons in the guest[12,13]. Three main categories of host materials exist based on their different charge transport properties: hole transporting (p-type), electron transporting (n-type), and ambipolar[14–16]. Regardless of which type of host is used, the triplet energy of the host should be higher than that of the TADF emitter to confine the singlet and triplet excitons on the emitter molecules[17].

In addition to a wide energy gap, hosts also need appropriate highest occupied molecular orbital (HOMO) and lowest unoccupied molecular orbital (LUMO) levels not only to balance the electron and hole fluxes but also to control the exciton formation mechanism[18,19]. There are two possible exciton formation mechanisms in an EML. In one, excitons form on the host molecules and then transfer their energy to the TADF emitters via the Förster or Dexter mechanism. In the second, electrons and holes directly combine on the TADF emitters to directly form excitons[20]. However, the latter mechanism seems more beneficial than the former for improving the efficiency and stability of TADF OLEDs[21]. Direct exciton formation on the TADF molecules can eliminate energy dissipation channels and avoid the formation of high-energy exciton on host molecules[22]. This is important because high-energy excitons can break the chemical bonds of TADF molecules to induce device degradation.

One challenge is that organic semiconductors typically show highly asymmetric hole and electron mobilities, with the hole mobility exceeding the electron mobility by orders of magnitude in most cases[23]. Thus, holes usually greatly outnumber electrons in the EMLs. These excess holes cannot recombine with electrons to form excitons and have a negative effect on the operational reliability of OLEDs because of interactions between hole-polarons and high-energy excitons[24,25]. Good charge balance in organic semiconductors can be achieved by either increasing the electron mobility or decreasing the hole mobility. Over the last three decades, many researchers have attempted to improve electron drift mobility, but it still lags far behind hole mobility[26,27].

Most TADF molecules consist of p-type (donor) and n-type (acceptor) moieties, leading to bipolar transport properties[28]. Unlike some stable fluorescent and phosphorescent emitters, the HOMO levels of TADF molecules are normally lower than −5.80 eV because of intrinsic properties (such as strong acceptor units and limited conjugation) of the emitters (Supplementary Table 3). Undoubtedly, the orbital energy levels of TADF emitters and host materials are critical for hole and electron transport channels in EMLs[29]. To realize charge balance and avoid high-energy exciton formation in EMLs, the ideal charge transport mode is hole transport on the TADF molecules and electron transport via the host molecules. To ensure that holes tend to be transported on the TADF molecules, the HOMO levels of the host molecules should be much deeper than those of the TADF molecules. Thus, we reasoned that n-type hosts are the preferred option for TADF emitters to improve both device efficiency and operational stability.

We tested our idea by designing and synthesizing three simple n-type hosts. Single-carrier current density–voltage ($J$–$V$) measurements clearly demonstrate that the hole and electron mobilities of these n-type hosts strongly depend on the TADF guest molecules. Such tuneable charge–carrier mobilities allow us to balance the hole and electron fluxes, broaden the exciton distribution and suppress the formation of high-energy exciton in EMLs. Through this strategy, we lengthen the lifetime of TADF OLEDs by > 30 times, revealing the possibility to achieve TADF OLEDs that are both efficient and stable.

## Results

**Characterization of n-type hosts.** The syntheses of our new n-type hosts SF2-TRZ, SF3-TRZ and SF4-TRZ are described in Supplementary Note 1. The three n-type hosts exhibit good thermal stability with clear glass transition temperatures above 130 °C (Supplementary Fig. 6a), and the decomposition temperatures at 5% loss are estimated to be nearly 400 °C (Supplementary Fig. 6b). The HOMOs of these isomers are located on the spirobifluorene (SF) group, whereas their LUMOs have slightly different distributions (Supplementary Fig. 1). Because of the large dihedral angle between the SF planes and triazine (TRZ) plane (36.3°) in an SF4-TRZ molecule, the LUMO of SF4-TRZ is only localized on the TRZ group and peripheral phenyl rings. Conversely, the dihedral angles completely vanished in SF2-TRZ and SF3-TRZ, and the LUMOs of both SF2-TRZ and SF3-TRZ spread over not only the TRZ group but also the SF unit. Spatially separated HOMOs and LUMOs (weak electron correlation) are beneficial for intramolecular charge transfer. Thus, the excited state of SF4-TRZ reveals charge transfer characteristics, which will be discussed in detailed later. The triplet spin density distributions (TSDDs) of these isomers were simulated to estimate their $T_1$ excited state locations. The TSDDs of SF2-TRZ and SF4-TRZ are mainly delocalized over SF and TRZ units, while that of SF3-TRZ is localized on the TRZ unit, suggesting a higher $T_1$ for SF3-TRZ (Fig. 1a).

Fig. 1b depicts the ultraviolet–visible (UV–vis) absorption and photoluminescence (PL) spectra of the three n-type hosts. The hosts present analogous absorption bands in the range of 304–315 nm with maxima at 309 nm. We attribute these bands, which are similar to the main absorption band of SF, to the $\pi$–$\pi^\star$ transitions of the fluorene fragment in SF. SF2-TRZ displays a strong absorption band at 326–370 nm, which is attributed to the extension of the $\pi$-conjugation length between the C2-substituted SF and TRZ units. Conversely, SF3-TRZ and SF4-TRZ show weaker absorption between 323 and 360 nm. This can be attributed to disruption of $\pi$-conjugation induced by the *meta*-linking mode and large torsion angle in SF3-TRZ and SF4-TRZ, respectively.

The PL spectra of the three analogs strongly depend on their substitution positions. SF2-TRZ exhibits a well-resolved emission spectrum ($\lambda_{max} = 403/417$ nm) that is a mirror image of its absorption spectrum. Conversely, SF3-TRZ and SF4-TRZ show structureless emission spectra with peaks at 412 and 448 nm, respectively. More importantly, compared with SF2-TRZ and SF3-TRZ, SF4-TRZ displays a broader emission spectrum. This indicates that SF4-TRZ displays charge transfer ($^1$CT) characteristics in its $S_1$ state, whereas the $S_1$ state of SF2-TRZ is identified as a localized state ($^1$LE) and that of SF3-TRZ might be composed of the mixture of a $^1$LE and a partial $^1$CT states. These differences

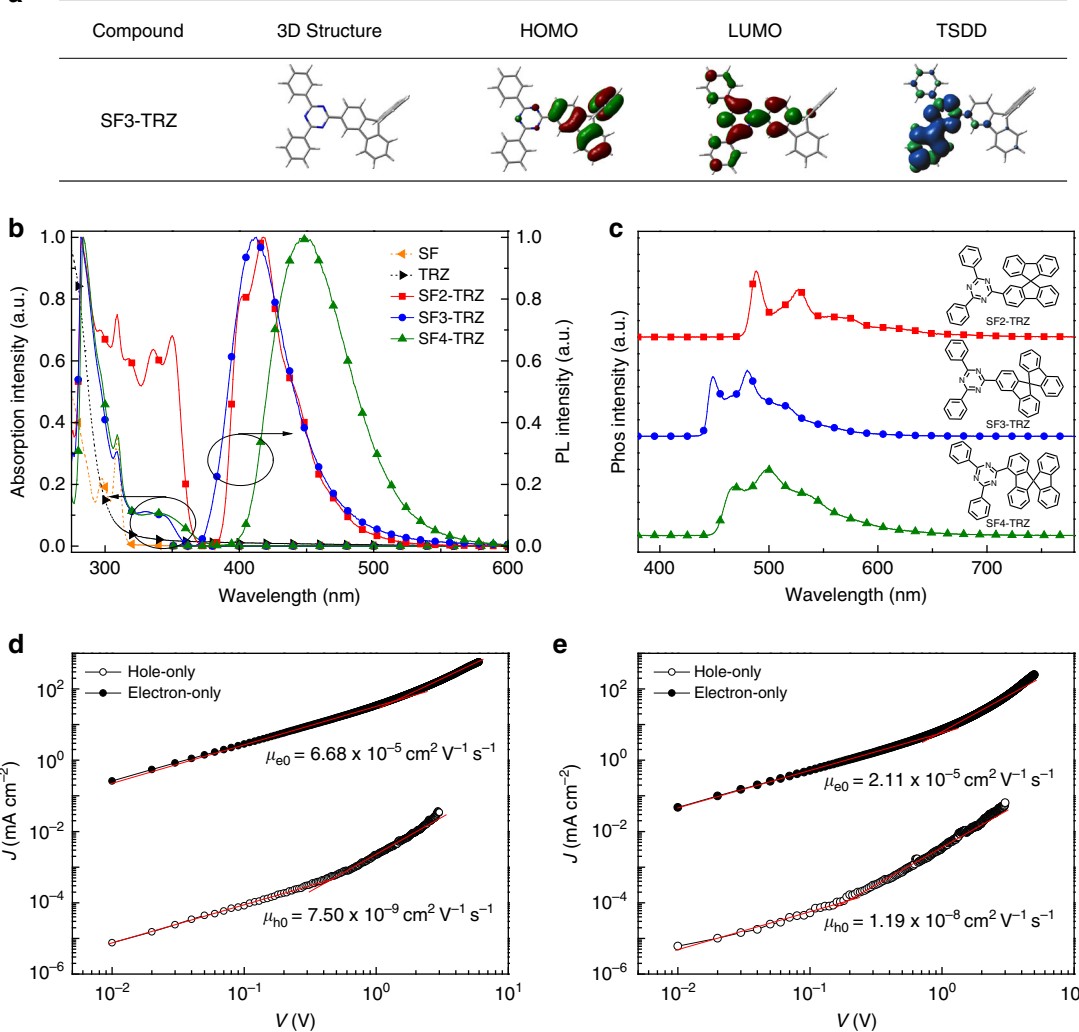

**Fig. 1** Molecular structures and properties. **a** Distribution of HOMO, LUMO and TSDD in SF3-TRZ. Optimized structures of the HOMO, LUMO, and TSDD of SF3-TRZ were calculated by TD-DFT (Gaussian09/B3LYP/6–31 G + (d)). **b** Absorption and photoluminescence spectra (298 K) of spirobifluorene (SF), triazine (TRZ), SF2-TRZ, SF3-TRZ and SF4-TRZ in dilute toluene solution. **c** Phosphorescent spectra of SF2-TRZ, SF3-TRZ, and SF4-TRZ in 2-methyltetrahydrofuran glass at 77 K. **d** Hole and electron transport in an SF2-TRZ neat film. Hole and electron current density (J) versus applied voltage (V) in an SF2-TRZ neat film. **e** Hole and electron transport in an SF4-TRZ neat film. Hole and electron J against V in an SF4-TRZ neat film

are well explained by the distributions of their frontier molecular orbitals (Supplementary Fig. 1). In addition, the emission peak of SF4-TRZ is red-shifted by ~36 nm compared with that of SF3-TRZ. This is because the $S_1$ state of SF4-TRZ undergoes substantial rigidification, such as a planarization of the structure, with the TRZ substituent conjugated to the SF core[30]. Thus, a bathochromic shift of emission and large Stokes shift are clearly observed for SF4-TRZ.

The solid-state UV–vis absorption and PL spectra of the three hosts (Supplementary Fig. 2) closely resemble those in solution state. This indicates that intermolecular interactions of these compounds are efficiently suppressed in their amorphous states because of their orthogonal molecular structures. The HOMO energy levels of SF2-TRZ, SF3-TRZ, and SF4-TRZ thin films were determined to be −6.56 eV, −6.54 eV, and −6.55 eV, respectively, using an AC-3 ultraviolet photoelectron spectrometer (Supplementary Figs. 3–5). The LUMO energy levels of SF2-TRZ, SF3-TRZ, and SF4-TRZ were estimated to be −3.27 eV, −3.10 eV, and −3.23 eV, respectively, by adding the optical energy gaps determined from the absorption edges of thin films to the HOMO energy levels. The $T_1$ energies of SF2-TRZ, SF3-TRZ, and

SF4-TRZ were determined to be 2.53, 2.80, and 2.65 eV, respectively, from the highest energy vibronic peak of their phosphorescent (Phos) spectra in 2-methyltetrahydrofuran (2-MeTHF) at 77 K (Fig. 1c). The high $T_1$ energy of SF3-TRZ makes it promising as a host for blue TADF emitters. The physical properties and energy levels are summarized in Supplementary Table 4.

The carrier transport properties of the n-type hosts were evaluated using hole-only devices (HODs) and electron-only devices (EODs) with structures of indium tin oxide (ITO)/MoO$_3$ (1 nm)/host (100 nm)/MoO$_3$ (10 nm)/Al (100 nm) and ITO/Cs (10 nm)/host (100 nm)/Cs (10 nm)/Al (100 nm), respectively. The J–V characteristics of the devices (Figs. 1d, e) can be divided into two distinct regions at low and high bias, which are assigned as the Schottky thermionic region and space-charge-limited current (SCLC) region, respectively. The SCLC region can be expressed by the equation

$$J = \frac{9}{8}\varepsilon\varepsilon_0\mu_0 \exp\left(\beta\sqrt{\frac{V}{L}}\right)\frac{V^2}{L^3} \qquad (1)$$

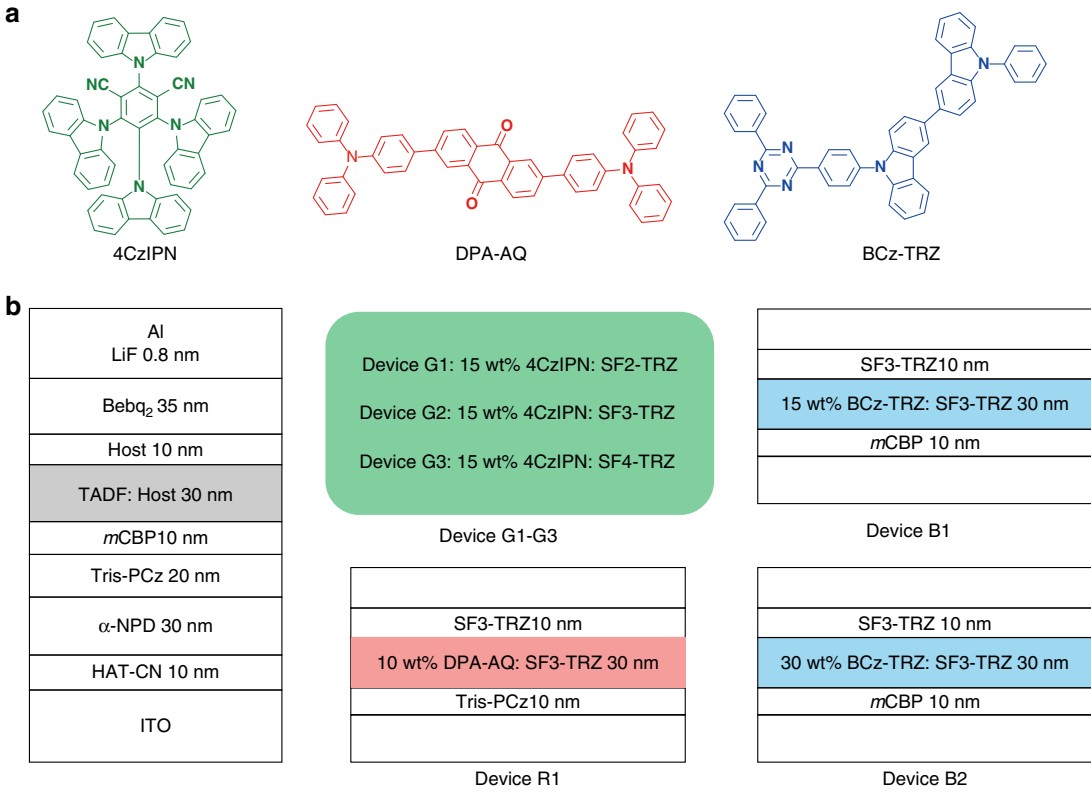

**Fig. 2** Structures of red, green and blue TADF OLEDs. **a** Chemical structures of the TADF emitters used in the emitting layers (EMLs) of TADF OLEDs. **b** Structures of TADF OLEDs

where $V$ is the applied voltage; $\mu_0$ is the zero-field charge mobility; $\varepsilon$ and $\varepsilon_0$ are the relative dielectric constant and free-space permittivity, respectively; $L$ is the thickness of the host material and $\beta$ is the Poole–Frenkel factor. Within this model, the mobility is expressed as

$$\mu = \mu_0 \exp\left(\beta\sqrt{\frac{V}{L}}\right) \qquad (2)$$

The estimated zero-field hole and electron mobilities ($\mu_{0h}$ and $\mu_{0e}$, respectively) of these n-type hosts are around $7–11 \times 10^{-9}$ and $2–6 \times 10^{-5}$ cm$^2$ V$^{-1}$ s$^{-1}$. The detailed model and data are summarized in Supplementary Note 2 and Supplementary Tables 6 and 7. These results agree well with the mobilities measured by the conventional time-of-flight method (Supplementary Fig. 9).

**TADF OLED performance**. We studied the electroluminescence (EL) properties of these n-type hosts in devices with the structure ITO/HAT-CN (10 nm)/α-NPD (30 nm)/Tris-PCz (20 nm)/$m$CBP (10 nm)/EML (30 nm)/host (10 nm)/Bebq$_2$ (35 nm)/LiF (0.8 nm)/ Al (120 nm), where HAT-CN, α-NPD, Tris-PCz, $m$CBP and Bebq$_2$ are 1,4,5,8,9,11-hexaazatriphenylene hexacarbonitrile, $N,N$ ′-diphenyl-$N,N$′-bis(1-naphthyl)−1,10-biphenyl-4,4′-diamine, 9,9′,9″-triphenyl-9$H$,9′$H$,9″$H$−3,3′:6′3″-tercarbazole, 3,3-di(9$H$-carbazol-9-yl)biphenyl and bis(10-hydroxybenzo[h]quinolinato) beryllium, respectively. EMLs consisting of the common TADF emitter 1,2,3,5-tetrakis (carbazol-9-yl)−4,6-dicyanobenzene (4CzIPN) doped in SF2-TRZ, SF3-TRZ and SF4-TRZ at an optimized doping concentration of 15 wt% were used in devices G1, G2 and G3, respectively, while the EML of device R1 consisted of 10 wt% 2,6-bis(4-(diphenylamino)phenyl)anthracene-9,10-dione (DPA-AQ) doped in SF3-TRZ (Fig. 2). For

comparison, a control device with the conventional p-type material $m$CBP as host was constructed.

As illustrated in Fig. 3a, high and stable efficiencies were achieved in devices G1, G2 and G3. The maximum efficiencies of device G2 were as high as 20.6% for external quantum efficiency (EQE), 68.3 cd A$^{-1}$ for current efficiency (CE), and 61.3 lm W$^{-1}$ for the power efficiency (PE). Device G3 exhibited moderate maximum efficiencies of 18.3% for EQE, 61.5 cd A$^{-1}$ for CE, and 54.3 lm W$^{-1}$ for PE. The efficiency of device G1 was lower than those of device G2 and G3, with maximum efficiencies being 14.5% for EQE, 50.1 cd A$^{-1}$ for CE, and 50.0 lm W$^{-1}$ for PE. The EQE values of devices G1, G2 and G3 follow the same trend as the PL quantum yields of films of 4CzIPN doped in the n-type hosts (Supplementary Table 5), which are determined by the $T_1$ energies of the hosts (Supplementary Table 4). In addition, all three devices exhibited lower efficiency roll-off at higher luminance compared to that of the $m$CBP-based device.

We evaluated the roll-off of the devices by comparing the critical luminance ($L_{90}$) at which the EQE decreases to 90% of its maximum value. The $L_{90}$ values of the SF-TRZ-based devices are as high as 3000 cd m$^{-2}$, which is much higher than that of the $m$CBP-based device (1700 cd/m$^2$, Supplementary Fig. 15), indicating a lower roll-off for the SF-TRZ-based devices. We ascribe this to the well-balanced transport of electrons and holes in the EMLs of device G1, G2 and G3 with n-type hosts, as described in detail in the following section.

The operational stability of the devices was also evaluated (Fig. 3d). The devices with the n-type hosts displayed long-term operational stability, with device lifetime extended more than 30 times compared to that of the device with a p-type host. Fig. 3d reveals that the half-lives ($T_{50}$) of devices G1, G2 and G3 are 565, 654 and 329 h for an initial brightness ($L_0$) of 5000 cd m$^{-2}$. The relatively short lifetime of device G3 may be caused by the weak bond dissociation energy between TRZ and SF moieties (bond C)

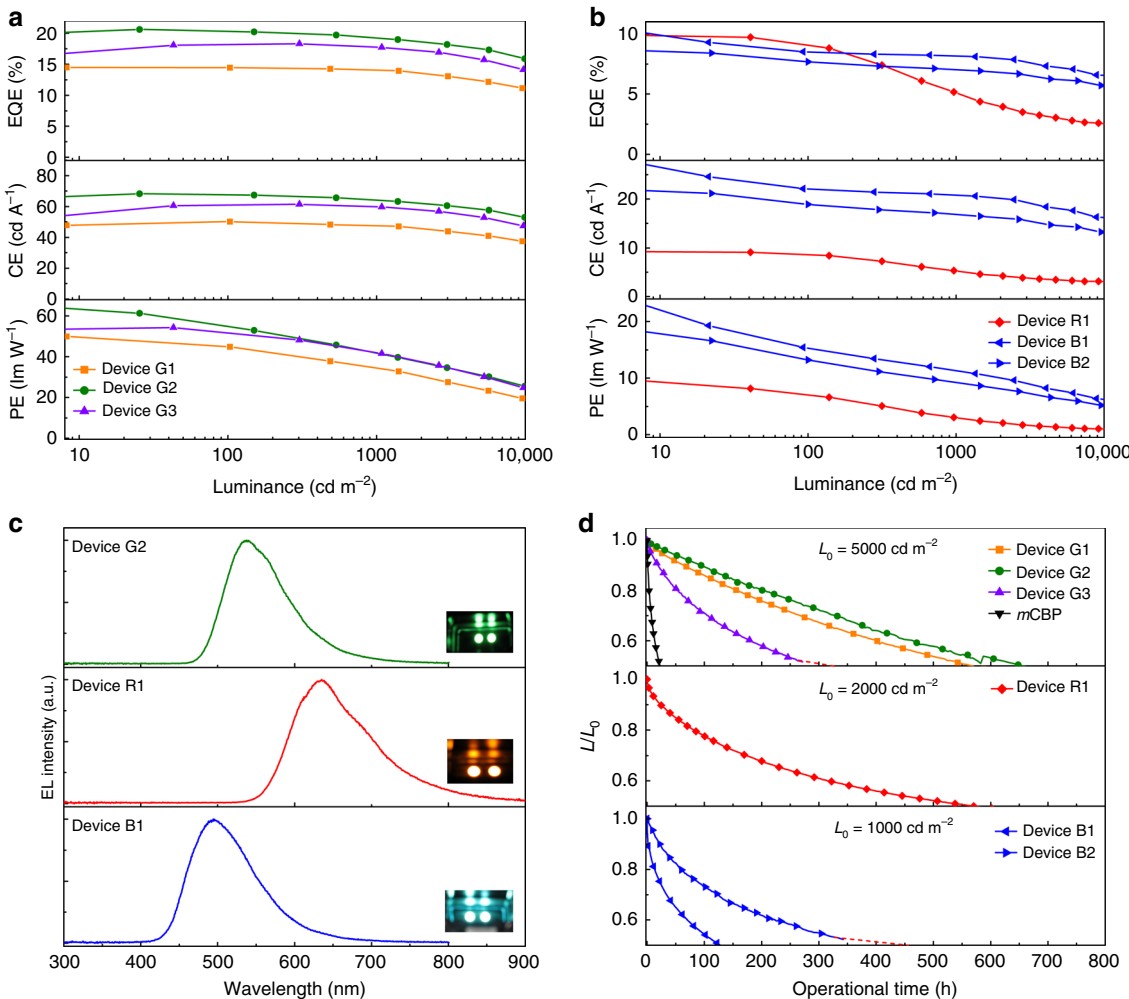

**Fig. 3** Performance characteristics of red, green and blue TADF OLEDs. **a** EQE, current efficiency (CE) and power efficiency (PE) versus luminance of devices G1, G2 and G3. **b** EQE, CE and PE versus luminance of devices R1, B1 and B2. **c** Electroluminescence (EL) spectra of device G2, R1 and B1 measured at a current density $J$ of 10 mA cm$^{-2}$. **d** Operational lifetime of the red, green and blue TADF OLEDs. The initial luminance of the green (G1–3), red (R1) and blue (B1 and 2) devices was 5000, 2000, and 1000 cd m$^{-2}$, respectively. The control device with $m$CBP as a host was operated at a constant current for an initial luminance of 5000 cd m$^{-2}$

in SF4-TRZ (Supplementary Table 2). The lifetime of device G2 is predicted to be 10,934 h at 1000 cd m$^{-2}$ according to the formula $LT(L_1) = LT(L_0)(L_0/L_1)^{1.75}$, where $L_1$ is the desired initial luminance of 1000 cd m$^{-2}$ [31].

Device R1 with a red emitter exhibits a high EQE of 11.5% and a $T_{50}$ of 594 h at an initial brightness of 2000 cd m$^{-2}$ (Figs 3b, d). The high $T_1$ of SF3-TRZ allowed it to be used as a host for blue TADF emitters. The EMLs of devices B1 and B2 respectively consisted of 15 and 30 wt% 9-(4-(4,6-diphenyl-1,3,5-triazin-2-yl) phenyl)-9′-phenyl-9$H$,9′$H$−3,3′-bicarbazole (BCz-TRZ) doped in SF3-TRZ. The EL spectra of devices G1, R1, and B1 are depicted in Fig. 3c. The maximum EQEs of device B1 and B2 were 11.0% and 8.8%, respectively (Fig. 3b). Importantly, the lifetime of device B1 was longer than that of device B2, with these devices displaying $T_{50}$ of 137 and 454 h, respectively, at an initial brightness of 1000 cd m$^{-2}$ (Fig. 3d). These values are amongst the longest reported for TADF OLEDs with such simple device architectures. The performance of all of the devices is summarized in Table 1.

Note that the lifetime of our control device in this study is shorter than that of our previous work[29]. The relatively short lifetime is mainly due to the different evaporation environment (all devices were fabricated under vacuum pressures below 3.0 ×

10$^{-4}$ Pa) and device architecture (excess holes deteriorate charge balance in our control device due to the replacement of the hole-transport material). In order to facilitate a straightforward comparison, we prepared the same device of our previous work under the same evaporation environment used in this study[29]. As shown in Supplementary Fig. 20, the device lifetime is slightly longer than that of our control device.

**Hole-only and electron-only devices**. We investigated HODs and EODs with 15 wt% 4CzIPN-doped n- and p-type hosts, respectively, to better understand the charge transport properties and exciton recombination zone in the EMLs (HOD: ITO/MoO$_3$ (1 nm)/host: 15 wt% 4CzIPN (100 nm)/MoO$_3$ (10 nm)/Al (100 nm); EOD: ITO/Cs (10 nm)/host: wt% 4CzIPN (100 nm)/Cs (10 nm)/ Al (100 nm)). Fig. 4 presents the $J$-$V$ characteristics of HODs and EODs with 15 wt% 4CzIPN-doped SF3-TRZ and $m$CBP layers. According to equation (1), doping 4CzIPN into SF3-TRZ leads to a more than thirtyfold increase in $\mu_{h0}$ of SF3-TRZ, whereas $\mu_{e0}$ of 4CzIPN-doped SF3-TRZ was nearly two orders of magnitude lower than that of SF3-TRZ. Therefore, the difference between the hole and electron mobilities of 4CzIPN-doped SF3-TRZ is relatively small. Conversely, $\mu_{h0}$ of 4CzIPN-doped $m$CBP is about

**Table 1 Summary of TADF OLED performance**

| Device | $V^a$ (V) | $EQE^b$ (%) | $CE^b$ (cd A$^{-1}$) | $PE^b$ (lm W$^{-1}$) | $T_{50}^c$ (h) | $CIE^d$ |
|---|---|---|---|---|---|---|
| G1 | 2.50, 3.48, 4.36 | 14.5, 14.4, 14.1 | 50.1, 50.0, 47.5 | 50.0, 45.0, 34.7 | 565 | (0.29, 0.58) |
| G2 | 2.54, 3.89, 4.85 | 20.6, 20.4, 19.2 | 68.3, 68.0, 64.0 | 61.3, 55.0, 42.1 | 654 | (0.29, 0.58) |
| G3 | 2.48, 3.76, 4.42 | 18.3, 18.2, 17.8 | 61.5, 60.9, 59.7 | 54.3, 51.4, 42.0 | 329 | (0.29, 0.58) |
| *m*CBP | 3.50, 5.33, 6.60 | 9.20, 9.10, 8.50 | 31.0, 30.4, 28.1 | 20.6, 17.9, 13.5 | 22.6 | (0.29, 0.58) |
| R1 | 2.49, 3.86, 5.54 | 11.5, 9.10, 5.20 | 10.1, 8.6, 5.30 | 12.6, 7.21, 3.12 | 594 | (0.63, 0.36) |
| B1 | 2.70, 4.53, 5.79 | 11.0, 8.5, 8.20 | 30.0, 22.1, 20.8 | 26.8, 15.3, 11.2 | 137 | (0.18, 0.34) |
| B2 | 2.62, 4.45, 5.75 | 8.80, 7.68, 7.02 | 22.5, 18.9, 16.8 | 20.1, 13.2, 9.20 | 454 | (0.20, 0.36) |

$^a$Operating voltage at onset, 100 cd m$^{-2}$, and 1000 cd m$^{-2}$
$^b$Values of external quantum efficiency (EQE) and current efficiency (CE) and power efficiency (PE) at their maximum, 100 cd m$^{-2}$, and 1000 cd m$^{-2}$
$^c$The operation lifetimes of green (G1–3 and *m*CBP-based device), red (R1), and blue (B1 and 2) devices were measured at an initial brightness of 5000, 2000, and 1000 cd m$^{-2}$, respectively
$^d$Recorded at 10 mA cm$^{-2}$

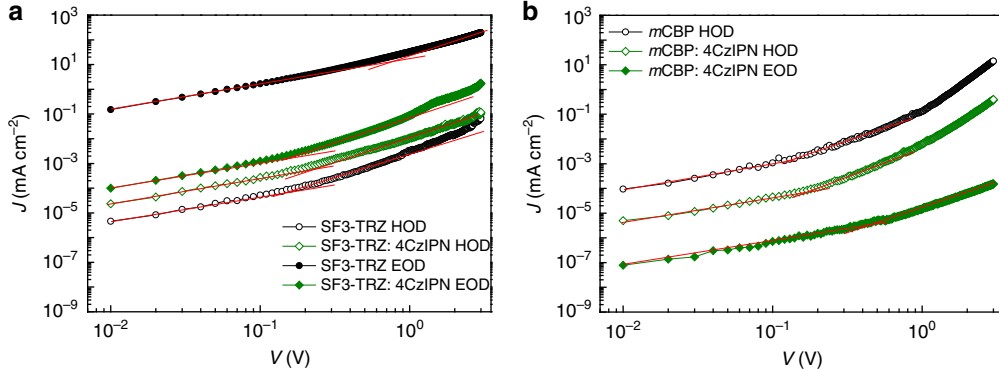

**Fig. 4** Hole and electron transport of host and TADF emitter-doped host films. **a** Hole and electron current density ($J$) versus applied voltage ($V$) in an SF3-TRZ neat film and 15 wt% 4CzIPN-doped SF3-TRZ film. **b** Hole and electron $J$ versus $V$ in an *m*CBP neat film and 15 wt% 4CzIPN-doped *m*CBP film

ten times lower than that of neat *m*CBP. Because its electron density was too low, $\mu_{e0}$ of a neat *m*CBP film could not be measured. However, the estimated $\mu_{e0}$ of the 4CzIPN-doped *m*CBP film was about $1.0 \times 10^{-10}$ cm$^2$ V$^{-1}$ s$^{-1}$. We believe that the electron mobility of *m*CBP should be increased by doping with 4CzIPN.

The difference between the hole and electron mobilities of 4CzIPN-doped *m*CBP is much larger than that of 4CzIPN-doped SF3-TRZ. Detailed data for the charge drift mobilities are listed in Supplementary Table 7. In addition, optical simulation (see Supplementary Note 3 for details) showed that, when using n-type hosts in the emitting layer, the recombination zone is broadened and shifted away from the cathode, resulting in a decrease of the Surface Plasmon Polariton (SPP) loss and an increase in the light outcoupling (Supplementary Fig. 21). Thus, we infer that use of n-type hosts in TADF OLEDs is greatly beneficial to balance the charge fluxes and subsequently broaden the recombination zone, avoid excessive charge and exciton accumulation, and lower leakage current in the devices. The negative effects on device stability induced by charge and exciton accumulation should be suppressed by using n-type hosts in TADF OLEDs. We further verified our hypothesis by fabricating HODs and EODs based on our EL device architectures. The differences between $J$ of the HODs and EODs based on n-type host SF3-TRZ are much smaller than those of ones based on p-type host *m*CBP at the same voltage (Supplementary Figs. 12 and 13).

## Discussion

Our observations can be well explained from the perspective of the different hole and electron transport channels in the EMLs caused by the difference between the orbital energy levels of host molecules and TADF emitters. TADF materials typically possess both hole and electron transport properties because of their intrinsic molecular structures. The HOMO levels of TADF emitters are usually deeper than those of fluorescent and phosphorescent emitters. For example, the HOMO level of 4CzIPN is −5.80 eV, which is close to that of the p-type host *m*CBP. Thus, holes tend to be transported on host molecules in a 4CzIPN-doped *m*CBP film. Although the electron mobility of *m*CBP can be improved by doping, electron transport is difficult to balance with hole transport because of their intrinsic asymmetry. In contrast, holes tend to be transported on TADF molecules in a 4CzIPN-doped SF3-TRZ film because of the deep HOMO level of the n-type host. This approach can increase the hole mobilities of n-type hosts, although TADF molecules do simultaneously lower the electron mobilities of n-type hosts through the dilution effect. Overall, the difference between the hole and electron mobilities of n-type host-based EMLs is much smaller than that of EMLs with a p-type host, indicating superior charge-carrier balance and a broad recombination zone in EMLs with an n-type host. In addition, the orbital levels of n-type hosts usually encourage exciton formation on TADF molecules, which can suppress high-energy exciton formation on host molecules and further extend device lifetime. These advantages are the most important reasons why n-type hosts are beneficial to improve the stability of TADF devices. However, the electron mobilities of EMLs based on p-type hosts are much lower than their hole mobilities. Therefore, hole and electron fluxes are unbalanced, resulting in exciton accumulation at the interface of the EML. By employing n-type hosts for TADF OLEDs, we successfully manipulated the hole and electron fluxes and substantially improve device operational stability, and this strategy has the advantages of simplifying the device architectures and reducing the manufacturing cost compared with the conventional mixed host strategy[32]. Above all, n-type hosts are proved to be the best choice for TADF OLEDs.

In summary, TADF molecules usually consist of donor and acceptor moieties, and the acceptor unit determines the HOMO level of a TADF emitter, which is typically deeper than −5.80 eV. This situation is quite different from that of the emitters in some stable fluorescent and phosphorescent OLEDs. Thus, efficient and stable TADF OLEDs require strict criteria regarding host selection to match with TADF emitters. Fortunately, n-type hosts possess inherent advantages to balance charge fluxes and suppress high-energy exciton formation because of their deep HOMO levels and excellent electron transport properties. Here we demonstrated a thirtyfold increase in the lifetime of TADF OLEDs upon using n-type hosts. Green TADF OLEDs with SF3-TRZ as the host achieved a maximum EQE of 20.6% and predicted $T_{50}$ of 10,934 h for an initial brightness of 1000 cd m$^{-2}$. More importantly, SF3-TRZ can also function as a host for sky-blue TADF OLEDs because of its high $T_1$. A sky-blue TADF OLED with a high EQE of 8.8% and lifetime of 454 h for an initial brightness of 1000 cd m$^{-2}$ was produced. This lifetime is three orders of magnitude higher than the device hosted by a p-type material CzSi (9-(4-tert-butylphenyl)−3,6-bis(triphenylsilyl)−9H-carbazole) according to our previous report[24]. Although the lifetimes reported here lag behind the criteria for consumer electronics, further lifetime improvements should be achieved by finding the most suitable host/dopant combinations. Our work offers guidelines to realize long-lived and efficient TADF OLEDs.

## Methods

**Materials**. All chemicals and materials, unless otherwise indicated, were purchased from Aldrich, Xi'an Polymer Light Technology Co. or Luminescence Technology Co. and used without additional purification. The molecular structures of the as-synthesized TADF molecules were fully characterized by NMR spectroscopy, mass spectrometry and elemental analysis (Supplementary Figs. 22–27).

**Quantum chemical calculations**. The software Gaussian 09 was used to perform all of the calculations. The geometries in the ground state were optimized via DFT calculations at the B3LYP/6−31 G + (d) level. TD-DFT calculations for the $S_0 \rightarrow S_n$ and $S_0 \rightarrow T_1$ transitions using the B3LYP functional were then performed according to the geometry optimization in the lowest-lying singlet and triplet states, respectively. Triplet spin density distributions (TSDDs) of unpaired electrons in the triplet state for SF2-TRZ, SF3-TRZ, and SF4-TRZ were characterized using Mulliken population analysis. The $\alpha$ and $\beta$ spin density distributions are indicated in blue and green, respectively, and the radius of the circle of each atom corresponds to the value of the TSDD. The calculated enthalpy change in the homolytic cleavage reaction of the corresponding single bond in the gas phase at 298 K and 1 atm was used to estimate bond dissociation energy.

**Photoluminescence measurements**. The absorption and photoluminescence characteristics of the hosts in the solution state were investigated in toluene solutions containing the three hosts ($1 \times 10^{-6}$ mol L$^{-1}$). Neat film samples were deposited on quartz glass substrates by vacuum evaporation to study their excitons confinement properties in the film state. UV–vis and PL spectra were measured using a Perkin-Elmer Lambda 950 KPA spectrophotometer and a JobinYvon FluoroMax-3 fluorospectrophotometer, respectively. Phosphorescent spectra were measured using a JASCO FP-6500 fluorescence spectrophotometer at 77 K. Absolute photoluminescence quantum yield was measured under nitrogen flow with excitation at 360 nm using a Quantaurus-QY measurement system (C11347−11, Hamamatsu Photonics). The prompt and delayed PL spectra of the samples under vacuum were measured using a streak camera system (Hamamatsu Photonics, C4334) equipped with a cryostat (Iwatani, GASESCRT-006-2000, Japan), and 337 nm light from a nitrogen gas laser (Lasertechnik Berlin, MNL200) was used as the excitation source. A UV photoelectron emission spectrometer (Riken Keiki AC-3) was used determine the HOMO energy levels of the compounds in the neat films.

**Device fabrication and evaluation**. OLEDs were fabricated by vacuum depositing the materials at ca. $3.0 \times 10^{-4}$ Pa onto ITO-coated glass substrates (sheet resistance of ca. 15 $\Omega/\square$). Before device fabrication, the ITO-coated glass substrates were cleaned in sequential ultrasonic baths of acetone, ethanol, and deionized water, dried in an oven, and then exposed to UV/ozone for about 30 min. After depositing the organic layers (deposition rates of 2–3 Å s$^{-1}$), the devices were unloaded into a nitrogen-filled glovebox and affixed to metal masks that defined the cathode area. The devices were loaded back into the same evaporation chamber for deposition of a cathode of LiF (0.2 Å s$^{-1}$) and Al (ca. 4 Å s$^{-1}$). The emitting area of all the OLEDs was determined by the overlap of the two electrodes (0.04 cm$^2$). The $J$–$V$–$L$ characteristics were measured using a Keithley 2400 source meter in conjunction with an absolute EQE measurement system (C9920−12, Hamamatsu Photonics, Japan).

**Optical simulations**. Optical simulations are performed using SETFOS 4.5 software in order to extract the recombination zone profile and position and to calculate the light outcoupling as well as the optical channel losses in the SF3-TRZ (n-type host) and mCBP (p-type host)-based OLEDs. The profile and the position of the recombination zone in the devices were extracted by fitting the measured EL spectra to an optical model. In this optical model the excitons are modeled as isotropic radiative dipoles driven by the multiple reflections inside the devices. The power radiated from a dipole at a certain wavelength is weighed by the photo-luminescence spectrum of the emitting layer. Using the same approach cited above, the dissipated power as a function of the wavelength and the in-plane wave vector can be calculated. The contribution of the modes is obtained by integrating the dissipated power.

**Data availability**. The data that support the findings of this study are available from the authors upon request.

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

## Acknowledgements

This work was financially supported by the Exploratory Research for Advanced Technology (ERATO) under JST ERATO Grant Number JPMJER1305, Japan, and the Natural Science Foundation of China (No. 61575136). L-.S.C. was supported by a scholarship from the China Scholarship Council (CSC). We thank W. J. Potscavage Jr for assistance with preparation of this manuscript.

## Author contributions

C.A. proposed and supervised the project. L-.S.C. designed the molecules and carried out experiments. S-.B.R and R.N. fabricated and characterized EODs and HODs. F.B. carried out the optical simulations. K.I. performed the computational experiments. L.Z. and L.S. L. provided the synthetic intermediates. H.N. assisted in manuscript preparation. L-.S.C. and C.A. analyzed the data and wrote the manuscript. All authors discussed the progress of research and reviewed the manuscript.

## Additional information

**Competing interests:** The authors declare no competing financial interests.

