## [Peer Review File · Nature Communications]

Reviewers' comments:

Reviewer #1 (Remarks to the Author):

The paper by Cui et al presents a series of new n-type organic host materials for TADF OLEDs. The materials all possess deep HOMO levels and higher electron mobilities than common p-type hosts, such as mCBP, which have previously been employed by the Adachi group.

The premise for using such materials is that of the available methods for reducing triplet-charge and triplet-triplet interactions, reducing the excess charge density (holes, usually) is a very promising approach. The transport measurements presented support the use of the SF-TRZ family in this role, and devices with good efficiencies and promising operational lifetimes are reported. The achievement of good operational lifetime for TADF materials is perhaps the most critical topic in the field.

The description of the materials, the origin of their luminescence and relative state energies are well presented and represent useful data for interpreting the device performance and host:guest PLQE. The associated calculations appear reasonable and help in this interpretation. The measured S1/T1 energies should be included in table S1, or similar, for easier comparison.

The difficulty with the paper as written is that considerable weight is placed on the importance of balancing charge mobilities, which is based on limited evidence. The effect is likely to be real, but the evidence presented is not particularly strong. For instance, it's not obvious that SCLC mobility measurements conducted (presumably) in the absence of blocking layers necessarily capture the limiting processes in a complete device. The current densities observed for bipolar devices are very different to those of the single-carrier devices. The structures of the HODs and EODs should be given so that this can be interpreted correctly, and the contribution of injection barriers ruled out. For instance, do sweeps both at forward and reverse bias yield the same mobility? What built-in voltage is assumed?

More importantly, the performance of the SF-TRZ devices is presented in a very favourable light compared to the p-type host mCBP and the associated control device. However, this device does not appear to be typical – while the comparison is not straightforward, both the EQE and lifetime compare poorly with devices of similar construction previously prepared by the Adachi group (e.g. doi:10.1038/srep02127). The T90 @ 1000cdm⁻² previously reported was over 200 hours, so a comparison to this value (scaled by luminance) is warranted.

The L90 values for the control and SF-TRZ hosts are also not very different. A factor < 2 for a measurement which depends heavily on the position of the maximum value should not be over weighted. In their prior publication a 15%-doped mCBP host exhibits L90 greater than 5000cdm⁻². The authors state that switching to an n-type host broadens the recombination zone, avoids excessive charge and exciton accumulation, and lowers leakage current. This is conjecture, since no data is presented to substantiate these claims. A broadening of the recombination zone should lead to a change in outcoupling profile, which will be detectable (see e.g. the work of Tessler et al). An accumulation of excitons or charge could be detected optically as a new absorption, and a lower leakage current should I think be detectable in the existing data (e.g. S13). Can the authors rule out a change in the triplet lifetime as a contributing factor?

Some more minor considerations:

P4 l69: This statement needs clarification, what intrinsic properties are referred to? Is this more than an empirical observation?

P5 I98: It's clearly the case that spatially separated HOMO/LUMO favours charge transfer, but why is this of benefit in general? Is this in reference to the exchange splitting for these hosts? The point should be clarified.

P5 I012: T1 is used to mean both the state and its energy, these should be distinguished.

P8 I177: Is the relation between PLQE and T1 energy quantifiable? Does the emission lifetime change, for instance? The supplementary notes refer to prompt and delayed fluorescence, but none is presented.

Overall, the authors present an interesting and reasonable approach to improving efficiency and stability in TADF OLEDs, and present data which suggests the mechanism by which this improvement occurs. However, overly-favourable comparison with a single sub-optimal control and a lack of supporting data weakens the argument. The authors' claims may well be correct, and the concept is sound, but in my opinion more rigor is required to conclude that this approach is more promising/simpler than utilising mixed-host systems, and the paper as presented needs revision to reflect this.

Reviewer #2 (Remarks to the Author):

This paper is a very valuable paper that solves the TADF lifetime problem well.

The paper is a well-written and a very interesting paper for many related researchers. Therefore, I recommend this paper to publish in Nature Communications.

However, in the case of the blue TADF material, the lifetime problems shows a lot of improvement but the efficiency and color purity are not good. Deep blue and efficient TADF dopants should also be tested for effectiveness. (For example, deep TADF blue dopant was reported by authors's group. *Angewante Chemie*, 2017, 56, 1571-1575 by C. Adachi et.al) In addition, it is necessary to add a table that summarizes the efficiency, color coordinates, and lifetime of each device.

Here we summarized our reply. First of all, we would like to thank the referees for providing us the valuable comments to improve this manuscript.

Answers to the reviewer #1

Question 1: The description of the materials, the origin of their luminescence and relative state energies are well presented and represent useful data for interpreting the device performance and host:guest PLQE. The associated calculations appear reasonable and help in this interpretation. The measured S_1/T_1 energies should be included in table S1, or similar, for easier comparison.

Answer: The physical properties of the three n-type hosts have been summarized in Table S4.

Question 2: The difficulty with the paper as written is that considerable weight is placed on the importance of balancing charge mobilities, which is based on limited evidence. The effect is likely to be real, but the evidence presented is not particularly strong. For instance, it's not obvious that SCLC mobility measurements conducted (presumably) in the absence of blocking layers necessarily capture the limiting processes in a complete device. The current densities observed for bipolar devices are very different to those of the single-carrier devices. The structures of the HODs and EODs should be given so that this can be interpreted correctly, and the contribution of injection barriers ruled out. For instance, do sweeps both at forward and reverse bias yield the same mobility? What built-in voltage is assumed?

Answer: Firstly, we fabricated HODs and EODs with 15 wt% 4CzIPN-doped n- and p-type hosts to investigate hole and electron mobilities of EMLs (HODs: ITO/MoO₃ (1 nm)/host: 15 wt% 4CzIPN (100 nm)/MoO₃ (10 nm)/Al (100 nm); EODs: ITO/Cs (10 nm)/host: wt% 4CzIPN (100 nm)/Cs (10 nm)/Al (100 nm)). Obviously, the hole and electron mobilities of host materials are dramatically affected by the TADF molecules. Interestingly, n- and p-type hosts exhibit different variation tendencies in their hole and electron mobilities (see Figure 4). The difference between the hole and electron mobilities of 4CzIPN-doped n-type host is much smaller than that of 4CzIPN-doped p-type host. It is

really difficult to deduce that the n-type host shows better charge carrier balance in a complete device based on this data because of absence of blocking layers in HODs and EODs as mentioned by reviewer. Thus, we further fabricated HODs and EODs based on our OLED architectures (HODs: ITO/ HAT-CN (10 nm)/ α -NPD (30 nm)/ Tris-PCz (20nm)/ mCBP (10 nm)/Host: 15 wt% TADF (30 nm)/ Tris-PCz (20 nm)/Al (100 nm); EODs: ITO/SF3-TRZ (20 nm)/ SF3-TRZ: 15 wt% TADF (30 nm)/ SF3-TRZ (10 nm)/ Bebq₂ (35nm)/LiF (0.8 nm)/Al (120 nm)). Obviously, the differences between current density of the HODs and EODs based on n-type host SF3-TRZ are much smaller than those of ones based on p-type host *m*CBP at the same voltage (see Supplementary Figures S11 and S12). Thus, we can infer that use of n-type hosts in TADF OLEDs is greatly beneficial to balance the charge fluxes and subsequently broaden the recombination zone. We supplemented the *J*-*V* characteristics of SF3-TRZ HOD (ITO/MoO₃ (1 nm)/SF3-TRZ (100 nm)/MoO₃ (10 nm)/Al (100 nm) and EOD (ITO/Cs (10 nm)/SF3-TRZ (100 nm)/Cs (10 nm)/Al (100 nm)) in forward and reverse bias directions (see Supplementary Figure S10). The HOD and EOD show almost symmetric *J*-*V* characteristics in the forward and reverse biases, indicating that the influence of the built-in potential on the *J*-*V* characteristics is completely negligible by using the MoO₃ and Cs as buffer layers in HODs and EODs.

Figure S10. *J*-*V* characteristics of SF3-TRZ single-carrier devices. **a**, *J*-*V* characteristics of electron-only device (ITO/Cs (10 nm)/SF3-TRZ (100 nm)/Cs (10 nm)/Al (100 nm)) in forward and reverse bias directions. **b**, *J*-*V* characteristics of hole-only device (ITO/MoO₃ (1 nm)/SF3-TRZ (100 nm)/MoO₃ (10 nm)/Al (100 nm)) in forward and reverse bias directions.

Question 3: More importantly, the performance of the SF-TRZ devices is presented in a very favourable light compared to the p-type host mCBP and the associated control device. However, this device does not appear to be typical – while the comparison is not straightforward, both the EQE and lifetime compare poorly with devices of similar construction previously prepared by the Adachi group (e.g. doi:10.1038/srep02127). The T90 @ 1000cdm-2 previously reported was over 200 hours, so a comparison to this value (scaled by luminance) is warranted. The L_{90} values for the control and SF-TRZ hosts are also not very different. A factor < 2 for a measurement which depends heavily on the position of the maximum value should not be over weighted. In their prior publication a 15%-doped mCBP host exhibits L_{90} greater than 5000cd/m².

Answer : Actually, the devices of this work and our previous work (doi:10.1038/srep02127) were fabricated in different evaporation environment. As described in Supplementary Information, all devices of this work were fabricated under vacuum at pressures below 3.0×10^{-4} Pa, whereas the vacuum pressures of our previous work (doi:10.1038/srep02127) can achieve 5.0×10^{-5} Pa. Besides that, the organic layers and metal electrodes in this work were deposited in the same evaporation chamber, while the organic layers were deposited in organic system evaporation chamber and metal electrodes were deposited in metal system evaporation chamber in our previous work (doi:10.1038/srep02127). Based on our recent work (DOI: 10.1038/srep38482), evaporation environment (such as vacuum pressure and chamber purity) plays a significant role in the device operational stability. We believe the above-mentioned issues are the main reasons that the lifetime of our control device in this work is shorter than that of our previous work (doi:10.1038/srep02127). In order to further confirm our explanations, we fabricated the same device of our previous work (doi:10.1038/srep02127) under the same evaporation environment used in this study. The device exhibited comparable efficiency with our previous work (doi:10.1038/srep02127), but the device lifetime is shorter (see Supplementary Figure S20). On the other hand, we used α -NPD as hole transport layer in this work to replace Tris-PCz in our previous work (doi:10.1038/srep02127). α -NPD ($\mu_0 = 7.64 \times 10^{-4}$ cm²/Vs; HOMO = -5.40 eV) has higher hole mobility and better hole injection

than Tris-PCz ($\mu_0 = 2.78 \times 10^{-6} \text{ cm}^2/\text{Vs}$; HOMO = -5.60 eV). Naturally, this leads to charge unbalance in the control device of this work, and resulting in serious efficiency roll-off and even inferior device lifetime. Thus, the deteriorated charge balance is considered to be the main cause of the lower L_{90} value the control device in this work.

Figure S20. Performance of OLED with structure ITO/HAT-CN (10 nm)/Tris-PCz (30 nm)/mCBP (10 nm)/mCBP: 15 wt% 4CzIPN (30 nm)/T2T (10 nm)/BPy-TP2 (35 nm)/LiF (0.8 nm)/Al (120 nm). **a**, J - V -luminance (L) characteristics. **b**, EQE versus luminance. **c**, Power efficiency (PE) and current efficiency (CE) versus luminance. **d**, Normalized EL intensity and voltage rise as a function of operational time at an initial luminance of 5,000 cd/m^2 . (Note that the device was fabricated under vacuum at pressures below 3.0×10^{-4} Pa and all layers (organic layers and metal electrodes) were deposited in the same evaporation chamber).

Question 4: The authors state that switching to an n-type host broadens the recombination zone, avoids excessive charge and exciton accumulation, and lowers leakage current. This is conjecture, since no data is presented to substantiate these claims. A broadening of the recombination zone should lead to a change in outcoupling profile, which will be

detectable (see e.g. the work of Tessler et al). An accumulation of excitons or charge could be detected optically as a new absorption, and a lower leakage current should I think be detectable in the existing data (e.g. S13). Can the authors rule out a change in the triplet lifetime as a contributing factor?

Answer: In order to further substantiate our claims, we simulated optical model using SETFOS 4.5 software to extract the recombination zone profile and position and to calculate the light outcoupling as well as the optical channel losses in the SF3-TRZ (n-type host) and *m*CBP (p-type host)-based OLEDs. The extracted recombination zone position and shape for *m*CBP and SF3-based OLEDs are shown in Supplementary Figures S21b and S21d, respectively. When a p-type host material (*m*CBP) is used, the recombination zone is situated at the cathode side (see Supplementary Figure S21b). Interestingly, when the n-type host material (SF3-TRZ) is used, the recombination zone is shifted to the other side away from the cathode due to the faster electron mobility (see Supplementary Figure S21d). In addition, the profile of the recombination zone is obviously wider in the case of the n-type-based.

The shift of the recombination zone affects the light trapping and outcoupling of the devices. In order to quantify the light trapping and outcoupling, the dissipated power as a function of the wavelength and the in-plane wave vector is calculated and shown in Supplementary Figures S21e and S21f for *m*CBP and SF3-based OLEDs, respectively. The different optical loss channels are clearly identified for both devices. The percentage of the optical power coupled to the different optical channels in the *m*CBP-based OLED and the SF3-based OLED are shown in Supplementary Figures S21g and S21h, respectively. In the *m*CBP-based OLED, the Surface Plasmon Polariton (SPP) loss is dominant (see Supplementary Figure S21e) and reach 69% whereas in the SF3-based OLED, due to the shift of the recombination zone away from the cathode, the SPP loss is decreased to 12% and the waveguide mode becomes dominant (see Supplementary Figure S21f). In addition, the substrate mode and the external light outcoupling are enhanced. The outcoupled light in the SF3-based OLED is found to be 19% against 9% for the *m*CBP-based OLED which is consistent with the experimental values.

Figure S21. Optical simulations of TADF OLEDs based on p-type and n-type host. **a** and **c**, Simulated and experimental EL spectra for *m*CBP-based OLED and SF3-based OLED, respectively. **b** and **d**, Recombination zone (Radiative dipole) distribution and position in *m*CBP-based OLED and SF3-based OLED, respectively. **e** and **f**, Calculated total dissipated optical power density for *m*CBP-based OLED and SF3-based OLED, respectively. **g** and **h**, Amount of the optical power coupled to the different optical channels in *m*CBP-based OLED and SF3-based OLED, respectively.

For triplet lifetime, we supplemented the transient PL decays of 15% 4CzIPN doped in SF3-TRZ and *m*CBP. Obviously, the two doped films exhibited the exactly same triplet lifetime (see Supplementary Figure S7). Thus, we could exclude the potential influence of triplet lifetime on the device operational stability.

Figure S7. Transient PL decays of *m*CBP and SF3-TRZ films doped with 15wt % 4CzIPN.

Some more minor considerations:

Question 5: This statement needs clarification, what intrinsic properties are referred to? Is this more than an empirical observation?

Answer: Typically, TADF molecules simultaneously contain strong donor and acceptor units. Most importantly, the π -conjugation between donor and acceptor units is usually limited in TADF molecules, which is well domesticated in our recent publication (10.1002/ange.201609459). The famous donor groups for TADF molecules are carbazole (HOMO \approx -5.9 eV), acridine (HOMO \approx -5.6 eV) and triphenylamine (HOMO \approx -5.6 eV) units. Although the HOMO levels of TADF molecules are mainly determined by donor units, acceptor groups in TADF molecules can also decrease the HOMO levels. Ultimately, the HOMO levels of TADF molecules are normally lower than -5.80 eV. However, we are

not saying that all TADF molecules have the HOMO levels lower than -5.80 eV and some might be exceptional. This is beyond our research. Therefore, we simply add the sentence if “strong acceptor units, limited conjugation and so on” after “intrinsic properties”.

Question 5: It's clearly the case that spatially separated HOMO/LUMO favours charge transfer, but why is this of benefit in general? Is this in reference to the exchange splitting for these hosts? The point should be clarified.

Answer: Our ambiguous expression may cause the reviewer misunderstanding. Actually, we would like to reflect that the singlet excited state (S_1) of SF4-TRZ is a charge transfer state. As shown in Figure 1b, the PL spectrum of SF4-TRZ is broad and structureless, which can further confirm this. From calculation (Figure S1), the HOMO and LUMO distributions of SF4-TRZ are well separated. Thus, electron correlations between HOMO and LUMO are very weak, which is beneficial to form charge transfer state SF4-TRZ. To obviate this misunderstanding, we add “weak electron correlation” after “Spatially separated HOMOs and LUMOs” and a sentence “Thus, the excited state of SF4-TRZ maybe reveals charge transfer characteristics, which will be discussed in detailed below” after “intramolecular charge transfer”.

Question 6: T_1 is used to mean both the state and its energy, these should be distinguished.

Answer: We have corrected this information.

Question 7: Is the relation between PLQE and T_1 energy quantifiable? Does the emission lifetime change, for instance? The supplementary notes refer to prompt and delayed fluorescence, but none is presented.

Answer: We supplemented the transient PL decays of 15% 4CzIPN doped in SF-TRZ series hosts (see Supplementary Figure S8). The 15% 4CzIPN: SF3-TRZ co-deposited film has the longest delayed lifetime ($3.65 \mu\text{s}$), which is attributable to the high triplet energy of SF3-TRZ. For relation between PLQE and T_1 energy, please find the detailed information from our previous papers (doi: [org/10.1016/j.orgel.2014.05.027](https://doi.org/10.1016/j.orgel.2014.05.027) and doi: [10.1038/srep02127](https://doi.org/10.1038/srep02127)).

Figure S8. Transient PL decays of SF2-TRZ, SF3-TRZ and SF4-TRZ films doped with 15wt % 4CzIPN.

Question 8: The authors' claims may well be correct, and the concept is sound, but in my opinion more rigor is required to conclude that this approach is more promising/simpler than utilizing mixed-host systems, and the paper as presented needs revision to reflect this.

Answer: We added a sentence “The key feature of our concept is employing n-type hosts for TADF OLEDs to manipulate hole and electron fluxes and then substantially improve device operational stability, which has advantages of simplifying the device architectures and reducing the manufacturing cost compared with the conventional mixed host strategy” in the discussion section.

Answers to the reviewer #2

Question : This paper is a very valuable paper that solves the TADF lifetime problem well. The paper is a well-written and a very interesting paper for many related researchers. Therefore, I recommend this paper to publish in Nature Communications. However, in the case of the blue TADF material, the lifetime problems shows a lot of

improvement but the efficiency and color purity are not good. Deep blue and efficient TADF dopants should also be tested for effectiveness. (For example, deep TADF blue dopant was reported by authors's group. *Angewante Chemie*, 2017, 56, 1571-1575 by C. Adachi et.al) In addition, it is necessary to add a table that summarizes the efficiency, color coordinates, and lifetime of each device.

Answer: Actually, the triplet energies of these n-type hosts are not high enough for deep-blue TADF emitters. Thus, high triplet energy n-type host materials should be explored. Fortunately, we have already obtained an n-type host with the triplet energy as high as 3.0 eV. We believe the long-lived deep-blue TADF OLEDs will be achieved in the near future. In addition, as the reviewer suggested, we added Table 1 to summarize the efficiency, color coordinates, and lifetime of each device.

List of Changes

Manuscript

Text

Pg. 1: a new co-author “Fatima Bencheikh” was added in the author line.

Pg. 4: the sentence “strong acceptor units, limited conjugation and so on” was added after “lower than -5.80 eV because of intrinsic properties”.

Pg. 5: the words “weak electron correlation” was added after “Spatially separated HOMOs and LUMOs”.

Pg. 5: the sentence “Thus, the excited state of SF4-TRZ maybe reveals charge transfer characteristics, which will be discussed in detailed below” was added after “beneficial for intramolecular charge transfer”.

Pg. 6: the sentence “More importantly, compared with SF2-TRZ and SF3-TRZ, SF4-TRZ displays a broader emission spectrum” was added after “show structureless emission spectra with peaks at 412 and 448 nm, respectively”.

Pg. 6: the sentence “SF3-TRZ maybe reveals partial charge transfer properties” was added after “the S_1 state of SF2-TRZ is identified as a localized state (1LE)”.

Pg. 7: the sentence “All the physical property data is summarized in Supplementary Table S4.” was added after “The high T_1 energy of SF3-TRZ makes it promising as a host for blue TADF emitters”.

Pg. 10: the sentence “All the device performances are summarized in Table 1” was added after “TADF OLEDs with such simple device architectures”.

Pg. 10: “(HOD: ITO/MoO3 (1 nm)/host: 15 wt% 4CzIPN (100 nm)/MoO3 (10 nm)/Al (100 nm); EOD: ITO/Cs (10 nm)/host: wt% 4CzIPN (100 nm)/Cs (10 nm)/Al (100 nm)).” was added after “HODs and EODs with 15 wt% 4CzIPN-doped n- and p-type hosts, respectively, were fabricated”.

Pg. 11: the sentence “In addition, optical simulation showed that when using n-type hosts material in the emitting layer, the recombination zone is broadened and shifted away from the cathode resulting in a decrease of the Surface Plasmon Polariton (SPP) loss and an increase in the light outcoupling (Supplementary Figure S21)” was added after “Detailed data for the charge drift mobilities are listed in Table S7”.

Pg. 12: the sentence “The key feature of our concept is employing n-type hosts for TADF OLEDs to manipulate hole and electron fluxes and then substantially improve device operational stability, which has advantages of simplifying the device architectures and reducing the manufacturing cost compared with the conventional mixed host strategy” was added after “Therefore, hole and electron fluxes are unbalanced, resulting in exciton accumulation at the interface of the EML”.

Pg. 15: “Optical simulations are performed using SETFOS 4.5 software (Fluxim AG) in order to extract the recombination zone profile and position and to calculate the light outcoupling as well as the optical channel losses in the SF3-TRZ (n-type host) and *m*CBP (p-Type host)-based OLED” was added in **Device fabrication and measurements** section.

References

Ref. 31: added the paper “Lee, J. H. *et al.* Mixed host organic light-emitting devices with low driving voltage and long lifetime. *Appl. Phys. Lett.* **86**, 103506 (2005)”

Table

Table 1: Summary of TADF OLED performances.

Supporting Information

Figure S7: Transient PL decays of *m*CBP and SF3-TRZ films doped with 15wt% 4CzIPN.

Figure S8: Transient PL decays of SF2-TRZ, SF3-TRZ and SF4-TRZ films doped with 15wt% 4CzIPN.

Figure S10: *J–V* characteristics of SF3-TRZ single-carrier devices. **a**, *J–V* characteristics of electron-only device (ITO/Cs (10 nm)/SF3-TRZ (100 nm)/Cs (10 nm)/Al (100 nm)) in forward and reverse bias directions. **b**, *J–V* characteristics of hole-only device (ITO/MoO₃ (1 nm)/SF3-TRZ (100 nm)/MoO₃ (10 nm)/Al (100 nm) in forward and reverse bias directions.

Figure S20: Performance of OLED with structure ITO/HAT-CN (10 nm)/Tris-PCz (30 nm)/*m*CBP (10 nm)/*m*CBP: 15 wt% 4CzIPN (30 nm)/T2T (10 nm)/BPy-TP2 (35 nm)/LiF (0.8 nm)/Al (120 nm). **a**, *J–V*–luminance (*L*) characteristics. **b**, EQE versus luminance. **c**, Power efficiency (PE) and current efficiency (CE) versus luminance. **d**, Normalized EL intensity and voltage rise as a function of operational time at an initial luminance of 5,000 cd/m².

Figure S21: Optical Simulations of TADF OLEDs based on p-type and n-type host. **a** and **c**, Simulated and experimental EL spectra for *m*CBP-based OLED and SF3-based OLED, respectively. **b** and **d**, Recombination zone (Radiative dipole) distribution and position in *m*CBP-based OLED and SF3-based OLED, respectively. **e** and **f**, Calculated total dissipated optical power density for *m*CBP-based OLED and SF3-based OLED, respectively. **g** and **h**, Amount of the optical power coupled to the different optical channels in *m*CBP-based OLED and SF3-based OLED, respectively.

Table

Table S4: Physical Properties of SF2-TRZ, SF3-TRZ and SF4-TRZ.

REVIEWERS' COMMENTS:

Reviewer #1 (Remarks to the Author):

The authors have now conducted additional experimental investigations and included further data which supports their conclusion. The paper is improved as a result.

My query regarding triplet energies has been addressed, additional data regarding the question of charge balance has been added, which addresses the point as far as is reasonable to request.

The authors have clarified the comparison of efficiencies and lifetimes with previous work, and conducted additional experiments which lend weight to their claims. The comparison is reasonable and their conclusions appear broadly justified.

The simulations of recombination zone position are useful and instructive, and the inclusion of triplet lifetime is helpful. The suggestion regarding excited state density has not been taken up but would likely require specialist equipment to establish. It is therefore reasonable to omit this data.

The clarifications requested have been made, and the authors points are now clearer. Their conclusions appear reasonable given the data available.

I recommend that the paper be published without further revision.

Reviewer #2 (Remarks to the Author):

Accept